# Cultural Competence and Cultural Intelligence of Healthcare Professionals Providing Emergency Medical Services

**DOI:** 10.3390/ijerph182111547

**Published:** 2021-11-03

**Authors:** Anna Majda, Iwona Elżbieta Bodys-Cupak, Joanna Zalewska-Puchała, Krystian Barzykowski

**Affiliations:** 1Laboratory of Theory and Fundamentals of Nursing, Institute of Nursing and Midwifery, Faculty of Health Sciences, Jagiellonian Univeristy Medical College, ul. Michałowskiego 12, 31-126 Krakow, Poland; anna.majda@uj.edu.pl (A.M.); j.zalewska-puchala@uj.edu.pl (J.Z.-P.); 2Institute of Psychology, Jagiellonian University, ul. Ingardena 6, 30-060 Krakow, Poland; krystian.barzykowski@uj.edu.pl

**Keywords:** healthcare providers/healthcare professionals, emergency medical system, cultural competence, cultural intelligence

## Abstract

Background: There are more and more foreigners in Poland who become clients of the Polish healthcare system. They use, among others, emergency medical services provided by healthcare professionals: doctors, nurses, and paramedics. Skillful care for culturally different patients requires cultural competencies and cultural intelligence to ensure good quality of care and cultural safety. The study aimed to measure and assess the cultural competencies and cultural intelligence of medical professionals working in hospital emergency departments (HEDs) and hospital emergency rooms (HERs) in Małopolska, a region in southern Poland. Methods: The following questionnaires were used in the study: the Cross-Cultural Competence Inventory (CCCI), the Cultural Intelligence Scale (CQS), and Questionnaire on Attitudes Towards Culturally Divergent People. In total, 709 medical professionals participated in the study, including 363 nurses, 223 paramedics, and 123 doctors. Results: Cultural intelligence—the overall score and the scores on the metacognitive, cognitive, motivational, and behavioral subscales were significantly higher among HED and HER doctors. Cultural competencies—the overall score and the score on the cultural adaptation subscale were also significantly higher among HED and HER doctors. The CCCI and CQS scores were influenced by selected variables: taking care of and close interactions with representatives of other cultural circles; staying outside Poland for more than a month. Doctors were the group of medical professionals that were most tolerant and most positive towards people from other cultures. Conclusions: The research results confirm the positive impact of contact of medical professionals with people from other cultures on their cultural competencies and cultural intelligence. They indicate the need for training in acquiring cultural competencies and developing cultural intelligence, especially among nurses. They demonstrate the need to raise awareness among HED and HER medical professionals about issues in intercultural care and to increase diversity efforts, especially among nurses.

## 1. Introduction

More and more foreigners in Poland become clients of the Polish healthcare system and use, among others, emergency medical services, also known as emergency care. Hence, healthcare professionals such as doctors, nurses, and paramedics in hospital emergency departments (HEDs) and hospital emergency rooms (HERs) provide assistance to an increasing number of people from other countries and cultural environments who represent different beliefs, practices, customs, and value systems. The literature emphasizes that, in such situations, cultural competencies and cultural intelligence play an important role [1,2,3,4,5,6,7,8,9,10].

According to the estimates of the Central Statistical Office (GUS) of 31 December 2019, the number of foreigners living in Poland was 2,106,101 (including Ukrainians 1,351,418, Belarusians 105,404, Germans 77,073, Moldovans 37,338, Russians 37,030, Indians 33,107, Georgians 27,917, Vietnamese 27,386, Turks 25,049, Chinese 23,838) [11]. This is approximately 1 million foreigners more than in 2018. It is estimated that in the first two months of the COVID-19 pandemic, i.e., March and April 2020, the number of foreigners in Poland decreased by 223,000, i.e., by 10.1% of the figure as at the end of February [12]. Illegal migrants who are not covered by official statistics should be added to these estimates. Currently, almost 460,000 foreigners have valid residence permits in Poland. In 2020, this number increased by just over 34,000. The epidemiological situation and related travel restrictions encouraged foreigners to extend their stay in Poland. This was especially true of citizens of Ukraine and Belarus, who had previously used the possibility of temporary migrations to a greater extent. Out of 457,000 foreigners who had valid residence permits on 1 January 2021, the largest groups were as follows: Ukraine—244,200 people, Belarus—28,800 people, Germany—20,500 people, Russia—12,700 people, Vietnam—10,900 people, India—9900 people, Italy—8500 people, Georgia—7900 people, China—7100 people, and Great Britain—6600 people [13].

The National Health Fund (NHF) does not have nationwide data and does not keep complete statistics on the treatment of foreigners and their use of medical services in the Polish healthcare system, including units of the Polish Emergency Medical Services (PRM) system [14]. Furthermore, the costs of treating uninsured foreigners are unknown. The only migrants whose access to the healthcare system is monitored are refugees [15]. In the event of a threat to their health or life during their stay in Poland, foreign tourists and foreigners use HEDs and HERs. International agreements and national legal regulations impose an obligation on Poland to provide healthcare access to persons who are permanently or temporarily in Poland. The legal situation of foreigners varies. The Office for Foreigners (UdSC) organizes care for people who are waiting to be granted international protection; they stay in reception centers and undergo the refugee procedure. People working legally in Poland and their families have the same health insurance as Polish citizens. The situation of illegal migrants staying in Poland is much more difficult. They are not covered by health insurance, which means they face many issues when it comes to reimbursement of the cost of health services. However, Poland’s legal status in force provides every person with the possibility of obtaining help in life-threatening situations, even in the absence of insurance, until the life-threatening condition is resolved [16].

Polish Emergency Medical Services (PRM) is a system established in Poland in order to provide medical services to people experiencing a sudden health emergency. PRM provides medical rescue operations. The system consists of outgoing medical rescue teams (ZRM): S-type ambulances (specialized), P-type ambulances (basic), N-type ambulances (neonatal), T-type ambulances (transport), Helicopter Emergency Medical Service (HEMS), and hospital emergency departments (HEDs). HEDs are supported by hospital emergency rooms (HER). The PRM employs system doctors, paramedics, and system nurses [16].

Cultural difficulties that arise in the process of building relations between medical personnel and a foreign patient are greater, the lower the degree of cultural competence of the treating and being treated and a positive or negative attitude towards people who are culturally different [17]. Training PRM employees in the field of cultural competencies may bring benefits in the form of better understanding of culturally different patients and improved communication [16]. In the Polish-language literature, there are only a few studies and publications showing how to deal with culturally different patients, and these concern mainly nurses [18,19,20,21,22,23,24], but which may be used by other medical professionals. There is one book aimed directly at paramedics [16]. No information is available for PRM employees. Postgraduate courses in the care of culturally different patients are conducted infrequently and usually by non-governmental organizations.

Skillful treatment of patients who are different in terms of culture, ethnicity, or religion requires appropriate preparation. Medical professionals, including professionals providing services in the field of emergency medicine, should acquire and develop basic cultural competencies and cultural intelligence already during studies. Cultural competencies and cultural intelligence should be developed at the undergraduate and postgraduate level in three areas: affective, cognitive, and behavioral. The first area relates to overcoming stereotypes and the belief that someone’s worldview is the most important and building a positive image and attitude towards people from other cultures. The second area is knowledge concerning the influence of religion and culture on people’s behavior in times of health and illness problems. The third area relates to verbal and non-verbal communication skills [25]. Only since 2012 has it been obligatory for educational programs in Poland to take multicultural issues into account in the education of students of medical faculties [26,27,28].

Cultural competencies (intercultural competencies) are defined as a set of uniform attitudes, behaviors, and principles of medical professionals, thanks to which it is possible to work effectively with people from different cultures. Cultural intelligence is the ability of an individual to understand, correctly infer, function, manage, and deal with situations characteristic of cultural diversity.

There is a need to conduct research that provides information on the cultural competencies and cultural intelligence of healthcare professionals working in the PRM system who are the first to provide help to foreigners [16]. Using standardized research tools allows for a reliable and accurate assessment of cultural competencies and cultural intelligence, subjected to psychometric validation in Polish conditions (in the context of a different healthcare system and separate cultural diversity), in accordance with the accepted standards [29,30,31,32,33].

The study aimed to measure and assess the cultural competencies and cultural intelligence of medical professionals working in hospital emergency departments (HEDs) and hospital emergency rooms (HERs) in Małopolska, a region in southern Poland.

## 2. Materials and Methods

### 2.1. Design and Data

The researchers reviewed the cities and hospitals with HEDs and HERs in Małopolska, a region in southern Poland. A list of 31 centers, i.e., 21 HEDs and 10 HERs, was created. Written requests were sent to the management boards of these institutions asking for permission to conduct research; consent was obtained to conduct research in 19 HEDs and 6 HERs. The criterion for inclusion in the research was that the respondent had an employment contract with an HED or HER. The exclusion criteria were other forms of employment in HED, HER, or employment in medical rescue teams (EMS). A list was prepared of 1052 healthcare professionals who were employed in these facilities and met the above criteria: 308 doctors, 439 nurses, and 305 paramedics. During the recruitment period (September–December 2020), the researchers distributed three printed research tools to all 1052 healthcare professionals. Some of the respondents, i.e., 320 people, did not consent to participation in the research; some, i.e., 23 people, returned not fully completed research tools. Finally, fully completed research tools were analyzed from a sample of 709 people, which constitutes 67% of the entire possible research group: 123 doctors, 363 nurses, 223 paramedics.

### 2.2. Sample

A total of 709 individuals participated in the study (482 female, 227 male) aged 20–70 (*M* = 40.38, *SD* = 11.56; 1 participant did not indicate their age). All but two participants indicated Polish nationality. No incentive was offered for participation in the study. In general, the pool consisted of the three following groups of participants: (1) nurses (N = 363), (2) medics (N = 122), and (3) paramedics (N = 223). This pool was specifically used in the analyses of the CQS and the Positive/Negative Attitude Towards Culturally Divergent People Questionnaire. However, as recommended by the author of the original scale [34], for the CCCI analyses we excluded 330 participants who scored higher than 15 on the “Lie and Social Desirability” scale (ranging from 16 to 30). Therefore, the final sample for the CCCI analyses consisted of 378 participants (224 females, 134 males; 169 nurses, 84 medics, and 125 paramedics) aged 22–69 (*M* = 38.21, *SD* = 11.32). Finally, nurses had the most extensive work experience (measured by the number of work years: *M* = 22.18, *SD* = 11.33) compared to medics (*M* = 13.98, *SD* = 11.95) and paramedics (*M* = 9.15, *SD* = 9.44), *F*(2, 705) = 103.38, *p* = 0.001, *η*^2^ = 0.23. Most of the participants (45%) were recruited from small towns (population up to 20,000 people); 26% came from middle-sized cities (21,000 to 99,000 people); 29% were from big cities (more than 100,000 people).

### 2.3. Measurement Tool

The following research instruments were used in the present study:

*Cross-Cultural Competence Inventory (CCCI).* CCCI is a broad, multidimensional instrument for the detailed and comprehensive measurement of cultural competence. It also touches on deeper layers: attitudes related to cultural sensitivity/awareness, cultural skills, and the application and use of cultural knowledge. CCCI measures three aspects of cultural competence: cognitive, emotional, and behavioral. This is particularly important considering the fact that the most commonly used definition of cultural competence relates directly to three areas: knowledge—providing culturally specific information; skills—involving multicultural interventions; attitudes—cultural empathy, openness, curiosity, tolerance, lack of prejudice in interpersonal relationships, awareness of one’s own value system and its limitations, awareness of different perspectives, and hierarchies of values, norms, and patterns of behavior [35]. The instrument consists of 58 items relating to the following 6 subscales: 1. Cultural adaptation—18 items; 2. Self-presentation—4 items; 3. Ambiguity/uncertainty tolerance—11 items; 4. Determination—7 items; 5. Engagement—11 items; 6. Mission—7 items. The scale of lies and social approval, treated as a control scale assessing the need for social acceptance, included 5 items.

Answers were provided with the use of a 6-point Likert scale, where 6 meant “I strongly agree” and 1 meant “I strongly disagree”. Cultural competence helps achieve effective communication with people of different cultures. CCCI obtained satisfactory psychometric properties and reliability (internal consistency of Cronbach’s alpha 0.70 to 0.94) [34,36,37,38,39,40]. Similarly, satisfactory results were obtained in a Polish study (internal consistency of Cronbach’s alpha 0.83 to 0.86) [30,32].

*Cultural Intelligence Scale (CQS)*. CQS is a slightly different concept: it is not as deep as CCCI; it is more focused on examining knowledge, including meta-level knowledge; it is less focused on attitude (e.g., mindfulness) and behavior. Similarly to other types of intelligence, cultural intelligence is understood as the ability to adapt to the surrounding environment and to interpret unknown and ambiguous behavior. It is defined as the ability to function effectively in an environment that is culturally different from ours [41]. CQS consists of 20 items, the scope of which covers the following 4 subscales: 1. Metacognitive CQ; 2. Cognitive CQ; 3. Motivational CQ; and 4. Behavioral CQ. Answers were provided with the use of a 7-point Likert scale, where 7 meant “I strongly agree” and 1 “I strongly disagree”. CQS is characterized by good reliability indexes in the range of 0.70–0.86 [41,42,43,44,45,46,47,48,49] in international studies. Polish studies have shown that CQS also has satisfactory psychometric properties: it is characterized by high reliability (Cronbach’s alpha 0.94 to 0.95) and sufficient theoretical and criteria validity [31].

*The Positive/Negative Attitude Towards Culturally Divergent People Questionnaire*. This questionnaire was adapted from previous studies (Barzykowski et al., 2019a, 2019b) and consisted of questions relating to the two main research areas: (1) participants’ attitudes to and experience of interacting with people from diverse cultural backgrounds; (2) their attitudes towards refugees. For the former, participants were asked whether they (a) had lived abroad for at least 1 month (Yes/No); (b) have had close relationships with culturally diverse people (Yes/No); (c) had treated or cared for culturally diverse patients/clients in the past (Yes/No). Regarding the latter, participants were instructed to think of refugees coming to Poland and to answer whether they should be accepted by the Polish government and granted free education. We also asked about past attendance of any course/workshop/seminar oriented towards development of cross-cultural competencies/skills (Yes/No). The total summary of Yes answers in this area may be treated as an indicator of a strong positive attitude towards culturally divergent people.

### 2.4. Data Analysis

STATISTICA software (version 12.0; StatSoft Europe, Hamburg, Germany) was used for statistical analysis. Specifically, we conducted a series of one-way ANOVAs on the total score in the CCCI and CQS (also on total scores within each subscale of the questionnaires) to find differences between the studied groups (i.e., nurses, medics, paramedics) treated as a between-subjects variable. Finally, to further verify whether participants who demonstrate positive relationships with and/or a positive attitude towards foreign populations, minorities, and migrants perform higher on the CQS and CCCI (as might be theoretically expected), we used a series of two-way ANOVAs: total score of CQS and CCCI was used as an outcome variable; group (nurses, medics, paramedics) and attitude (e.g., positive close relationship vs. a lack of a positive close relationship with foreign individuals that were operationalized as answering Yes = positive close relationship, or No = the lack of positive close relationship, to the following question: *do you have a close relationship with any culturally diverse people*) were used as between-subjects factors. For all statistical analyses reported below, the level of significance was set at *p* < 0.05, and the effect size was measured by partial eta-squared (*ηp*^2^).

### 2.5. Ethical Considerations

Participants were provided with a document explaining the purpose of the study, showing the benefits, potential risk, and the possibility of voluntary withdrawal from the study. Participants were tested individually; they were informed that their anonymity would be protected and that they were free to withdraw from the study at any point. The study design received was approved by the Academic Bioethics Committee (No. KBET/1072.6120.222.2020). The study was developed and conducted in accordance with (1) the principles of Good Scientific Practice; (2) the Act of 10 May 2018, on the protection of personal data; (3) the principles of the Helsinki Declaration; and (4) the Regulation of the European Parliament, and of the Council (EU) 2016/679 of 27 April 2016, on the protection of individuals with regard to the processing of personal data. The study participants were provided with all the necessary information about the study, its purpose, and its procedure. Each participant received oral and written information about the purpose of the study and that participation was voluntary. Importantly, all participants were guaranteed the right to withdraw from participation in the study at any time without giving a reason or suffering any consequences. Submitting the completed questionnaires was tantamount to consent to participation in the study. Written consent for participation was obtained prior to data collection. In order to maintain anonymity, identification marks were provided on the research instruments. The data were anonymized by randomization, i.e., random separation of data to eliminate the close relationship between the data and a specific natural person, and coding, i.e., changing the values of variables from real to fictitious.

## 3. Results

### 3.1. Cross-Cultural Competence Inventory (CCCI)

The overall means of the CCCI (and for each subscale) across groups are provided in Table 1. As can be seen, nurses scored significantly worse on the CCCI compared to medics (*p* = 0.008). However, the differences between nurses and paramedics (*p* = 0.111) and between paramedics and medics (*p* = 0.238) were not statistically significant. As shown in Table 1, when looking at the subscales of the CCCI, we observed several significant group effects. Post hoc tests indicated that medics obtained the highest cultural adaptation scores compared to both groups: nurses (*p* = 0.002) and paramedics (*p* = 0.001). At the same time, paramedics scored the highest on the Determination and Tolerance scales compared to nurses (*p_Determination_* = 0.004, *p*_Tolerance_ = 0.001) and medics (*p_Determination_* = 0.028, *p*_Tolerance_ = 0.003). Finally, nurses scored lowest on the Self-presentation scale compared to both groups (*p*_Paramedics_ = 0.041, *p*_Medics_ = 0.045). We did not observe any significant group effect for the Mission and Engagement subscales.

### 3.2. Cultural Intelligence Scale (CQS)

The overall means for the CQS (and for each subscale) across groups are provided in Table 2. There was a significant group effect on the total score of the CQS as well as on all the CQS subscales. Specifically, post hoc tests indicated that medics compared to nurses and paramedics obtained the highest (1) CQS overall scores (*p*_Nurses_ = 0.001, *p*_Paramedics_ = 0.013), (2) Motivational CQ (*p*_Nurses_ = 0.001, *p*_Paramedics_ = 0.009), (3) Behavioral CQ (*p*_Nurses_ = 0.001, *p*_Paramedics_ = 0.027). At the same time, compared to paramedics and/or medics, nurses scored worse on (1) CQS overall scores (*p*_Paramedics_ = 0.001), (2) Metacognitive CQ (*p*_Medics_ = 0.008), (3) Cognitive CQ (*p*_Paramedics_ = 0.001), (3) Motivational CQ (*p*_Paramedics_ = 0.004).

### 3.3. The Relation between the Positive/Negative Attitude towards Culturally Divergent People Questionnaire and the CCCI and CQS across Groups

We analyzed the differences across groups in the overall CCCI and CQS scores between people who declared providing professional healthcare to culturally different people in two separate factorial ANOVAs: the overall score in the CCCI or CQS was used as an outcome variable; group (nurses, paramedics, medics) and past experience with professional healthcare of culturally diverse individuals (Yes vs. No) were used as between-subject factors. First, regarding CCCI, we observed significant main effects for (1) group, *F*(2, 372) = 3.98, *p* = 0.019, *ηp*^2^ = 0.01, and (2) having previous experience with culturally divergent healthcare, *F*(1, 372) = 9.99, *p* = 0.002, *ηp*^2^ = 0.03. A post hoc test indicated that nurses performed significantly lower on the CCCI scale compared to medics (*p* = 0.007) and individuals who had previous experience in taking care of culturally diverse people compared to those without such an experience scored significantly higher on the CCCI scale. We repeated similar analyses for the CQS. Similarly to the previous analyses, the main effects of group (*F*(2, 702) = 16.26, *p* = 0.001, *ηp*^2^ = 0.04) and of previous multicultural healthcare experience (*F*(1, 702) = 5.21, *p* = 0.023, *ηp*^2^ = 0.01) were statistically significant. Specifically, nurses scored significantly worse in the CQS compared to paramedics (*p* = 0.001) and medics (*p* = 0.001). At the same time, compared to individuals who did not declare previous cross-cultural healthcare experience, those who did have this experience scored significantly higher on the CQS (*p* = 0.026). The group by previous experience interaction was not statistically significant.

We repeated these analyses for individuals who had had close relationships with culturally diverse people and had lived abroad for at least 1 month. Those who fulfilled these criteria scored higher on both CCCI (*F*(1, 372) = 15.22, *p* = 0.001, *ηp*^2^ = 0.04) and CQS (*F*(1, 702) = 28.89, *p* = 0.001, *ηp*^2^ = 0.04). In addition, those who had lived abroad for at least 1 month also scored significantly higher on CCCI (*F*(1, 372) = 5.98, *p* = 0.015, *ηp*^2^ = 0.02) and CQS (*F*(1, 702) = 31.86, *p* = 0.001, *ηp*^2^ = 0.04). There were no significant group by the multicultural relationship interaction for any of these analyses. In none of the analyses was there any significant group in the multicultural relationship. As the majority of the participants did not declare any previous attendance of cross-cultural training/workshops (92%, 94%, 89% for nurses, paramedics, and medics, respectively), we were not able to analyze the differences between participants on the CQS and CCCI in this regard.

Finally, we analyzed the difference between groups in the total score on the Positive/Negative Attitude Towards Culturally Divergent People Questionnaire towards refuges (second area of the questionnaire). The main group effect was significant (*F*(2, 705) = 22.32, *p* = 0.001, *ηp*^2^ = 0.06). A post hoc test revealed that medics obtained the highest score on the positive attitude towards refuges (*M* = 0.94, *SD* = 0.83), compared to nurses (*p* = 0.001, *M* = 0.46, *SD* = 0.67) and paramedics (*p* = 0.001, *M* = 0.49, *SD* = 0.70).

## 4. Discussion

Cultural competencies (CC), also known as cultural intelligence (CQ) [50], are considered essential in clinical professions [51]. The golden rule is that we should treat patients in the same way we want others to treat us in times of vulnerability and fear. We do not need to memorize the social customs, prevailing beliefs, or rules associated with each culture in order to provide excellent care for people of other religions, ethnic groups, countries, and races. The key to developing cultural competencies and cultural intelligence is focusing on the patient and displaying respect, sensitivity, composure, partnership, honesty, acumen, curiosity, and tolerance, and a positive attitude towards them. All people want someone to care for them [52].

Cultural awareness is the ability of healthcare personnel to understand and respond to the unique cultural needs of HED and HER patients. According to the American College of Emergency Physicians, doctors should consider a patient’s culture in relation to their history and present symptoms. The treatment plan should contain elements of culture and be mutually agreed upon by the patient and the doctor. This organization also notes the dependence of healthcare quality on doctors’ scientific competencies and cultural awareness. Healthcare professionals should receive cultural awareness training in order to provide high-quality care in the emergency room. They should also encourage patients and their representatives to raise cultural issues that affect their care. Information resources should be made widely available to HED staff so that they can ensure that they appropriately respond to the needs of all patients, regardless of their origin [53].

Language barriers, socio-economic conditions, religious values, and cultural practices can be an obstacle to delivering high-quality care to an increasingly diverse patient population. These obstacles contribute to the difference in healthcare delivery. Improving cultural competencies is mentioned as part of the solution when it comes to reducing disparities. An emergency department is an environment in which cultural sensitivity is especially needed. It is often the primary healthcare facility for neglected ethnic and racial minorities. HED is a place where help is provided to a large number of patients in highly time-pressured conditions. The emergency medicine literature says little about differences in healthcare and cultural competencies. Padela and Punekar (2009) present three clinical scenarios that illustrate the challenges in providing equitable medical services to minority populations in the United States. One episode describes a Latino woman with shortness of breath; the second presents a Cambodian refugee with hemoptysis who does not speak English; the third describes a Muslim woman with a tingling left leg. Using these cases as an illustration, these authors propose three processes that can improve the quality of care provided to minorities: (a) increasing cultural awareness and reducing healthcare provider bias; (b) adjusting to patients’ preferences and needs in the field of medical procedures; (c) increasing the diversity of providers to increase the level of tolerance, awareness, and understanding of other cultures, and patient–doctor relationships that are more compatible with racially and/or ethnically diverse patients [1].

In Poland, patients from different cultural contexts are a new phenomenon. Therefore, medical professionals working in HED and HER may not have the experience, knowledge, skills, or culturally sensitive attitudes to deal with culturally different patients. Cooperation with patients during accident or illness is challenging due to the misfortune that affects them and the accompanying emotions and stress. A patient from a different culture may not speak Polish or English, and medical professionals usually do not know the patient’s native language. In such circumstances, it will be dependent on the awareness of medical professionals regarding interactions between people from different cultures, their sensitivity to the patient brought up in a different culture, and knowledge about a given culture whether the patient will find themself in a new situation, whether they will gain the patient’s trust and give him or her optimal medical assistance. Knowledge about behaviors related to the health of representatives of different cultures, their response to stress, their methods of communication, and their approach to using professional healthcare, including in emergency situations, is the basis for acquiring practical cultural competencies and cultural intelligence and using these when providing medical services to patients from different ethnic, national, or religious groups. Pre- and postgraduate education programs should include multicultural content. Medical students should be prepared theoretically and practically to help culturally different patients; therefore, education programs should teach attendees to take into account patients’ expectations that arise as a result of socio-cultural factors during medical rescue operations (MCR).

The analysis of the literature on the subject shows that intercultural competencies in Poland are measured among students of medicine and nursing as well as working doctors and nurses more often than among paramedics but not as often as in America or Western Europe [20,29,33,54]. Casillas et al. compared intercultural skills between doctors and nurses from the Swiss University Hospital in Lausanne. The questionnaire used for the study of cultural competencies consisted of questions from the Cross-Cultural Care Survey and questions written by the authors. Using a validated tool, the average scores of doctors and nurses and the percentages of “3—good, 4—very good” responses were compared for nine perceived skills (4-point Likert scale). Linear regression was used to investigate how the roles of healthcare professionals (doctors vs. nurses) were related to skill scores adjusted to demographic factors (gender, non-French language), workplace (duration of working in a hospital, individual “sensitive” to cultural care), reported training in the field of cultural competencies and awareness of the problems of intercultural care. The questionnaire was completed by 124 (33.6%) doctors and 244 (66.4%) nurses. Doctors had better mean scores for the observed skills and a much higher “good/very good” percentage of responses for 4/9 items. Among all respondents, lower cultural skills were associated with the perception/awareness of problems in inadequate intercultural training and a lack of practical care experience among a diverse population [55]. In the presented research, one wonders about the fact that nurses have low results in terms of Self-presentation and high scores on the Lie and Social Approval Scale (the strong need to show themselves from the good side). Self-presentation in CCCI is one of the important components of cultural competencies; it influences their overall indicator as it relates, among others, to the ability to be friendly despite an aversion to a person from a different culture. Results differing from those presented are visible in the studies by Szkup et al. [20], in which the youngest employees showed the greatest empathy towards people who are culturally different; the least empathy was observed in the 46–50 age group. Respondents aged 21–25 were most open to new experiences; respondents aged 36–45 were the least open. Women showed a higher level of ethnocultural empathy and openness to new experiences than men. Nurses showed the best ability to understand the feelings of a culturally different person and were also characterized by high cognitive flexibility [20].

In the presented research, people (most often doctors) with close relationships, experience in providing care to people from different cultural areas, and a positive attitude towards culturally different people obtained higher scores in terms of cultural intelligence and cultural competencies.

### 4.1. Limitations

The strength of this study is the diagnosis of cultural competencies and cultural intelligence among medical professionals in HEDs and HERs. The main limitation of this study is that it was conducted in one province that is relatively not very diversified in terms of religion and culture. Thus, the possibility of generalizing the results is limited. The hindering factors in this study could be, for example, the size and representativeness of the sample, the fact that the research was conducted during the COVID-19 epidemic, the impact of the social approval variable, because, as E. Zasępa writes after Karyłowski, it is saturated with tools measuring feelings and moral behavior [56] and the fact that few of the surveyed medical professionals took part in formal educational programs in the field of multiculturalism, as it was only since 2012 that these issues were obligatory in the education of medical students. The limitations and confounding factors that are typical of the applied methodology means that the results should be treated with caution. Future research could be extended to other places. It is recommended to repeat the study on an even larger sample.

### 4.2. Implications for Practice

The results may be helpful in practice and suggest that attention should be paid to improving the cultural competencies and cultural intelligence of medical staff in HEDs and HERs, changing attitudes towards culturally different people. They indicate that the content, methods, and forms of educational programs should be so designed and attractive as to change not only the knowledge, but also the skills and attitudes of medical professionals in the provision of culturally adequate care.

## 5. Conclusions

The research results confirm the positive impact of contact of medical professionals with people from other cultures on their cultural competencies and cultural intelligence. They indicate the need for training in acquiring cultural competencies and developing cultural intelligence, especially among nurses. They demonstrate the need to raise awareness among HED and HER medical professionals about issues in intercultural care and to increase diversity efforts, especially among nurses.

## Figures and Tables

**Table 1 ijerph-18-11547-t001:** Overall results of the Cross-Cultural Competence Inventory (CCCI) and subscales across groups.

	Nurses *M* (*SD*)	Paramedics *M* (*SD*)	Medics *M* (*SD*)	Test
**CCCI—overall result**	208.33 (23.48)	213.10 (27.28)	217.33 (26.16)	*F*(2, 375) = 3.74, *p* = 0.025, *ηp*^2^ = 0.02
**Cultural adaptation**	71.22 (13.84)	69.33 (14.90)	77.26 (14.57)	*F*(2, 375) = 8.05 *p* = 0.001, *ηp*^2^ = 0.04
**Determination**	24.87 (5.46)	26.74 (5.26)	25.02 (5.97)	*F*(2, 375) = 4.55 *p* = 0.011, *ηp*^2^ = 0.02
**Ambiguity/uncertainty tolerance**	30.60 (9.70)	35.22 (10.00)	31.08 (9.20)	*F*(2, 375) = 8.94 *p* = 0.001, *ηp*^2^ = 0.05
**Self-presentation**	10.42 (3.87)	11.38 (3.94)	11.49 (4.26)	*F*(2, 375) = 3.00 *p* = 0.051, *ηp*^2^ = 0.02
**Mission**	30.41 (6.33)	30.05 (6.42)	31.14 (5.73)	*F*(2, 375) = 0.78 *p* = 0.458, *ηp*^2^ = 0.01
**Engagement**	40.80 (8.41)	40.38 (8.33)	41.33 (8.36)	*F*(2, 375) = 0.32 *p* = 0.723, *ηp*^2^ = 0.01

*M*—mean, *SD*—standard deviation, CCCI—Cross-Cultural Competence Inventory.

**Table 2 ijerph-18-11547-t002:** Overall results of the Cultural Intelligence Scale (CQS) and subscales across groups.

	Nurses *M* (*SD*)	Paramedics *M* (*SD*)	Medics *M* (*SD*)	Test
**CQS—overall result**	66.83 (23.79)	73.49 (22.64)	79.94 (20.91)	*F*(2, 705) = 16.55 *p* = 0.001, *ηp*^2^ = 0.04
**Metacognitive CQ**	15.05 (5.57)	15.77 (5.54)	16.58 (5.14)	*F*(2, 705) = 3.82 *p* = 0.022, *ηp*^2^ = 0.01
**Cognitive CQ**	16.71 (7.96)	19.96 (8.33)	21.66 (6.62)	*F*(2, 705) = 23.26 *p* = 0.001, *ηp*^2^ = 0.06
**Motivational CQ**	17.19 (7.19)	18.87 (6.47)	20.87 (6.40)	*F*(2, 705) = 14.20 *p* = 0.001, *ηp*^2^ = 0.04
**Behavioral CQ**	17.88 (8.16)	18.90 (7.46)	20.84 (7.26)	*F*(2, 705) = 6.68 *p* = 0.001, *ηp*^2^ = 0.02

*M*—mean, *SD*—standard deviation, CQS—Cultural Intelligence Scale, CQ—cultural intelligence (quotient).

## Data Availability

The data presented in this study are available on request from the first author.

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
