# Peer review of "Cultural Competence and Cultural Intelligence of Healthcare Professionals Providing Emergency Medical Services"

_ijerph, 2021, doi:10.3390/ijerph182111547_

Round 1

Reviewer 1 Report

I recommend to authors :

  • Explain abbreviations and individual letters in the tables below the tables.  (p. 6)
  • In the references to the author's work, state the year of publication after the author's name (e.g p.8 Pedela and Punekar (2009) present....
  • Describe the Implications for practice more specifically. (p.9) 

Author Response

  • Explain abbreviations and individual letters in the tables below the tables.(p. 6)

Thank you, the abbreviations explained (p. 6), in the tables below the tables 1 and 2

Line 264

Table 1.

M – mean, SD - Standard Deviation, CCCI - Cross-Cultural Competence Inventory

Line 275-6

Table 2.

M – mean, SD - Standard Deviation, CQS - Cultural Intelligence Scale, CQ - cultural intelligence (quotient)

  • In the references to the author's work, state the year of publication after the author's name(g p.8 Pedela and Punekar (2009) present....

Thank you for your good attention, there has been a change (p. 8)

Line 344

Padela and Punekar (2009) present……..L

  • Describe the Implications for practice more specifically. (p.9)

Thank you, described implications for practice more specifically (p. 9)

The results may be helpful in practice and suggest that attention should be paid to improving the cultural competencies and cultural intelligence of medical staff in HEDs and HERs, changing attitudes towards culturally different people. They indicate that the content, methods and forms of educational programs should be so designed and attractive as to change not only the knowledge, but also the skills and attitudes of medical professionals in the provision of culturally adequate care. Lines 428-431

Reviewer 2 Report

Major points

  1. Introduction - previous work could be better reviewed and critically analyzed, e.g. what different are CCCI, CQS and the Positive/Negative Attitude Towards Culturally Divergent People Questionnaire between doctors, nurses, and paramedics? This will help the readers understand the context of the research and what to expect the different from healthcare professionals providing emergency medical services.

  1. Introduction - Added the novelty of your study compared to previous studies. For example, the theoretically expected of Positive/Negative Attitude Towards Culturally Divergent People should be extended.

  1. Discussion - The relationship between cultural awareness and CCCI, CQS and the Positive/Negative Attitude Towards Culturally Divergent People Questionnaire are not established in the introduction. Please confirm the meaning (line 320-330). The influences of the different of CCCI, CQS and the Positive/Negative Attitude Towards Culturally Divergent People Questionnaire between doctors, nurses, and paramedics should be discussed.

Minor points

  1. I think there are mistakes in line 252 “…Table 2” and in line 264 “…Table 3”.

  1. Since 2012, educational programs in Poland have begun to take multicultural issues into account in the education of students of medical faculties [26–28]. (line 106-108) The experiences of educational programs could be identified as a limitation in discussion section and future directions of research.

  1. The recommendations in implications for practices should be extended, e.g. the suggestive contents of education programs from the outcomes in this study.

Author Response

Major points

  1. Introduction - previous work could be better reviewed and critically analyzed, e.g. what different are CCCI, CQS and the Positive/Negative Attitude Towards Culturally Divergent People Questionnaire between doctors, nurses, and paramedics? This will help the readers understand the context of the research and what to expect the different from healthcare professionals providing emergency medical services.

Thank you, completed

Cultural difficulties that arise in the process of building relations between medical personnel and a foreign patient are the greater, the lower the degree of cultural competence of the treating and being treated and a positive or negative attitude towards people who are culturally different [17]. Lines 90-91

  1. Introduction - Added the novelty of your study compared to previous studies. For example, the theoretically expected of Positive/Negative Attitude Towards Culturally Divergent People should be extended.

 Thank you, completed

Cultural difficulties that arise in the process of building relations between medical personnel and a foreign patient are the greater, the lower the degree of cultural competence of the treating and being treated and a positive or negative attitude towards people who are culturally different [17]. Lines 90-91

The first area relates to overcoming stereotypes and the belief that someone's worldview is the most important and building a positive image and attitude towards people from other cultures. Line 106

The second area is knowledge concerning the influence of religion and culture on people's behavior in times of health and illness problems. The third area relates to verbal and non-verbal communication skills [25]. Only Ssince 2012, educational programs in Poland have begun obligatory to take multicultural issues into account in the education of students of medical faculties [26–28]. Lines 109-110

  1. Discussion - The relationship between cultural awareness and CCCI, CQS and the Positive/Negative Attitude Towards Culturally Divergent People Questionnaire are not established in the introduction. Please confirm the meaning (line 320-330). The influences of the different of CCCI, CQS and the Positive/Negative Attitude Towards Culturally Divergent People Questionnaire between doctors, nurses, and paramedics should be discussed.

Thank you, completed. The introduction was supplemented with an attitude. In the discussion, the results regarding the attitude were added.

Cultural difficulties that arise in the process of building relations between medical personnel and a foreign patient are the greater, the lower the degree of cultural competence of the treating and being treated and a positive or negative attitude towards people who are culturally different [17]. Lines 90-91

The first area relates to overcoming stereotypes and the belief that someone's worldview is the most important and building a positive image and attitude towards people from other cultures. The second….Lines 105-106

……..high cognitive flexibility [20]. In the presented research, people (most often doctors) with close relationships, experience in providing care to people from different cultural areas and a positive attitude towards culturally different people obtained higher scores in terms of cultural intelligence and cultural competences. Lines 406-409

Minor points

  1. I think there are mistakes in line 252 “…Table 2” and in line 264 “…Table 3”.

Thank you, a correction was made

Instead in line 252 Table 2 should Table 1   Line 254

Instead in line 264 Table 3 should Table 2   Line 267

  1. Since 2012, educational programs in Poland have begun to take multicultural issues into account in the education of students of medical faculties [26–28]. (line 106-108) The experiences of educational programs could be identified as a limitation in discussion section and future directions of research.

 Thank you, included in limitations

…….. and the fact that few of the surveyed medical professionals took part in formal educational programs in the field of multiculturalism, as it was only since 2012 that these issues were obligatorily in the education of medical students.…….. Lines 418-421

  1. The recommendations in implications for practices should be extended, e.g. the suggestive contents of education programs from the outcomes in this study.

Thank you, completed

The results may be helpful in practice and suggest that attention should be paid to improving the cultural competencies and cultural intelligence of medical staff in HEDs and HERs, changing attitudes towards culturally different people. They indicate that the content, methods and forms of educational programs should be so designed and attractive as to change not only the knowledge, but also the skills and attitudes of medical professionals in the provision of culturally adapted care. Lines 428-431

Reviewer 3 Report

Thank you for the opportunity to review the original article Cultural Competence and Cultural Intelligence of Healthcare Professionals Providing Emergency Medical Services

I appreciate the interesting and important topic of this study.

Comments and suggestions for the Authors:

The Abstract contains basic information, keywords are corresponding with the content of the manuscript.

I propose a few amendments in the Introduction section that could improve the quality of work.

Line 63

An erroneous abbreviation was used next to the name of the National Health Fund (NFZ), should be (NHF) – line 63

I propose to better organize information on the possibilities of educating health care workers in Poland.

  1. 1. “In the Polish-language literature, there are only a few studies and publications showing how to deal with culturally different patients, and these concern only nurses [18–24]. No information is available for PRM (Lines 92-95)

In my opinion, this knowledge is universal and can be used by all healthcare professionals regardless of the fact that the literature is addressed to nurses. In addition, there is a position on the market in Polish addressed to paramedics "A culturally different patient in the practice of a paramedic”. (Kiszka J., Pacjent odmienny kulturowo w praktyce ratownika medycznego, Wyd. UR 2020).

  1. The authors also point out that

 “Courses in the care of culturally different patients are conducted infrequently and usually by non-governmental organizations.” (Lines 95-96)

which is inconsistent with the information

„Since 2012, educational programs in Poland have begun to take multicultural issues into account in the education of students of medical faculties [26–28]. (Lines 106-108)

The methods are adequately described, their selection and execution are consistent with the type of empirical study.

Inclusion/exclusion criteria were clearly stated.

Results are well described.

The overall quality of the discussion sections is satisfactory. The authors also indicated work limitations.

The structure of the work complies with the requirements for the authors.

Author Response

Thank you for your positive feedback

Thank you for the opportunity to review the original article Cultural Competence and Cultural Intelligence of Healthcare Professionals Providing Emergency Medical Services. I appreciate the interesting and important topic of this study.

Comments and suggestions for the Authors:

The Abstract contains basic information, keywords are corresponding with the content of the manuscript.

I propose a few amendments in the Introduction section that could improve the quality of work.

Line 63

An erroneous abbreviation was used next to the name of the National Health Fund (NFZ), should be (NHF) – line 63

Thank you, changed to NHF  Line 63

I propose to better organize information on the possibilities of educating health care workers in Poland.

  1. 1. “In the Polish-language literature, there are only a few studies and publications showing how to deal with culturally different patients, and these concern only nurses [18–24]. No information is available for PRM ” (Lines 92-95)

In my opinion, this knowledge is universal and can be used by all healthcare professionals regardless of the fact that the literature is addressed to nurses. In addition, there is a position on the market in Polish addressed to paramedics "A culturally different patient in the practice of a paramedic”. (Kiszka J., Pacjent odmienny kulturowo w praktyce ratownika medycznego, Wyd. UR 2020).

Thank you, completed. The item indicated by the Reviewer 3 “A culturally different patient in the practice of a paramedic” was quoted under number [16] in the line 92.

Training PRM employees in the field of cultural competencies may bring benefits in the form of better understanding of culturally different patients and improved communication [16]. In the Polish-language literature, there are only a few studies and publications showing how to deal with culturally different patients, and these concern only mainly nurses [18–24], but which may used by other medical professionals. There is one book aimed directly at paramedics [16]. Lines 95-96

No information is available for PRM employees. Postgraduate cCourses in the care of culturally different patients are conducted infrequently and usually by non-governmental organizations. Line 97

  1. The authors also point out that

 “Courses in the care of culturally different patients are conducted infrequently and usually by non-governmental organizations.” (Lines 95-96)

which is inconsistent with the information

„Since 2012, educational programs in Poland have begun to take multicultural issues into account in the education of students of medical faculties [26–28]. (Lines 106-108)

Thank you, completed. Courses are voluntary and university education programs are compulsory. This is not contradictory information. They present the actual state of education in the field of cultural competences.

Only since 2012, educational programs in Poland have begun obligatory to take multicultural issues into account in the education of students of medical faculties [26–28]. Lines 109-110

Thank you for your positive feedback

The methods are adequately described, their selection and execution are consistent with the type of empirical study. Inclusion/exclusion criteria were clearly stated. Results are well described. The overall quality of the discussion sections is satisfactory. The authors also indicated work limitations. The structure of the work complies with the requirements for the authors.

Round 2

Reviewer 2 Report

Thanks for careful explaining and adding.